# The effect of mHealth on childhood vaccination in Africa: A systematic review and meta-analysis

**Girma Gilano**[1]*, **Sewunet Sako**[1ම], **Berihun Molla**[1ම], **Andre Dekker**[2‡], **Rianne Fijten**[2‡]

**1** Department of Public Health Informatics, School of Public Health, College of Medicine and Health Sciences, Arba Minch University, Arba Minch, Ethiopia, **2** Department of Radiation Oncology [Maastro], GROW School for Oncology and Developmental Biology, Maastricht University Medical Centre+, Maastricht, The Netherlands

ම These authors contributed equally to this work.
‡ AD and RF also contributed equally to this work.
* gilanog@yahoo.com

**Data Availability Statement:** All relevant data are within the paper and its Supporting Information files.

**Funding:** The author(s) received no specific funding for this work.

## Abstract

### Introduction

Vaccine-preventable diseases are the public health problems in Africa, although vaccination is an available, safe, simple, and effective method prevention. Technologies such as mHealth may provide maternal access to health information and support decisions on childhood vaccination. Many studies on the role of mHealth in vaccination decisions have been conducted in Africa, but the evidence needs to provide conclusive information to support mHealth introduction. This study provides essential information to assist planning and policy decisions regarding the use of mHealth for childhood vaccination.

### Methods

We conducted a systematic review and meta-analysis for studies applying mHealth in Africa for vaccination decisions following the Preferred Reporting Items for Systematic and Meta-Analysis [PRISMA] guideline. Databases such as CINAHL, EMBASE, PubMed, PsycINFO, Scopus, Web of Science, Google Scholar, Global Health, HINARI, and Cochrane Library were included. We screened studies in Endnote X20 and performed the analysis using Revman 5.4.1.

### Results

The database search yielded 1,365 articles [14 RCTs and 4 quasi-experiments] with 21,070 participants satisfied all eligibility criteria. The meta-analysis showed that mHealth has an OR of 2.15 [95% CI: 1.70–2.72; P<0.001; $I^2$ = 90%] on vaccination rates. The subgroup analysis showed that regional differences cause heterogeneity. Funnel plots and Harbord tests showed the absence of publication bias, while the GRADE scale showed a moderate-quality body of evidence.

**Competing interests:** The authors have declared that no competing interests exist.

**Abbreviations:** mHealth, Mobile Health; RD, Risk Difference; WHO, World Health Organization; UNICEF, United Nations International Child Emergency Fund; OR, Odds Ratio; USA, United States of America.

## Conclusion

Although heterogeneous, this systematic review and meta-analysis showed that the application of mHealth could potentially improve childhood vaccination in Africa. It increased childhood vaccination by more than double [2.15 times] among children whose mothers are motivated by mHealth services. MHealth is more effective in less developed regions and when an additional incentive party with the messaging system. However, it can be provided at a comparably low cost based on the development level of regions and can be established as a routine service in Africa.

## Registration

PROSPERO: CRD42023415956.

## Introduction

Vaccination is the safest, simplest, and most effective strategy to confer immunity against nearly 20 childhood diseases [1]. Vaccines signal the body to prepare against diseases before they come in contact with them and can save millions of lives [2].

As stipulated by the Centers for Disease Control and Prevention [CDC], childhood vaccination prevents 4 million deaths every year. This is shown by the global reduction of deaths due to vaccination-preventable diseases from 12.5 to 5.3 million from 1990 to 2018 [3]. Vaccination is expected to save 14 and 19 million children from Hepatitis B and measles, respectively. However, vaccination coverage is decreasing currently and was only 81% in 2021 throughout the globe [4]. Consequently, 25 million more children do not receive vaccination routinely, and 18 million receive zero doses [1, 2, 4]. Due to limited childhood vaccination, Africa has a high child mortality rate of 1 in 9 compared to 1 in 199 in developed countries [3].

According to the 2021 World Health Organization [WHO] report, the high child mortality in Africa is attributable to vaccine-preventable diseases in more than half of cases [5, 6]. The reasons for this are manifold, but people living in the Sub-Saharan region have high missed vaccination opportunities because of individual factors, for instance, sociodemographic variables, family's financial capacity, place of birth, and upbringing [7]. This means individuals and combined factors contribute to the low vaccination rates and associated high childhood mortality.

In recent years, there has been a rapid increase in applications of mHealth to increase the rate of childhood vaccination in many countries throughout the globe [8–10]. Africa is one of the continents where mHealth has been tested as an intervention to improve vaccination rates [9, 11, 12]. In Ethiopia, completeness and timeliness of vaccination were improved through text message reminders [12] and the number of revisits for vaccination increased in Tanzania because of Short Message Service [SMS] text reminders [13]. Additionally, a mobile phone-based platform improved the childhood vaccination rate in Kenya [14]. At the same time, another study in Nigeria showed an enhancement in the overall number of childhood vaccinations due to SMS reminders [9]. Furthermore, other studies showed an increased number of vaccination visits due to mHealth in Benin [15], Kenya [11, 16], Burkina Faso [17], Nigeria [18], Ghana [19], and Zimbabwe [20]. The above evidence shows reminders and vaccination information, which can be delivered through mobile phones, can potentially improve childhood vaccination [21].

Many challenges in Africa affect routine vaccinations and the application of mHealth [22, 23]. Africa is behind in the global vaccination rate for different reasons. For instance, maternal

education of any level [primary, secondary, or tertiary] [21], and political instability through civil war [24]. Dependency on unsustainable multinational foreign aid, which usually leads to unprecedented financial restraints [25], low uptake of vaccination in Muslim families, which represent a significant part of African populations [26], and lack of transportation to health centers in rural areas are some of the challenges in addition to economic factors and false beliefs regarding vaccines safety [27–29].

The mHealth-related challenges include poor usability, lack of system integration, poor data security and privacy, poor network access, and poor reliability [30]. In addition, other challenges such as low literacy, cultural constraints to accept mHealth, and healthcare-seeking behavior are prominent in Africa [15, 19, 31, 32]. Additional challenges are related to poor standardization, a challenging regulatory framework, and poor overall health system readiness [33]. These findings indicate that the implementation of mHealth in Africa needs multifactorial considerations.

Facilitators of childhood vaccination include positive clinical encounters, free vaccination service policy, optimal vaccine, and device supply chain system, adequate knowledge of vaccination benefits and efficacy, vaccination outreaches, and provision of incentives to caregivers [34, 35]. Another study identified early childhood vaccination facilitators' rural communities as a reminder/recall system and positive parents' relationships with providers [36].

In conclusion, there are many challenges and opportunities regarding the application of mHealth to improve childhood vaccination. In the presence of challenges, the existing evidence is inconclusive and inconsistent if mHealth can improve childhood vaccination in challenging African contexts. For this, we summarized barriers and facilitators of mHealth to improve childhood vaccination and provided conclusive evidence. The summarized evidence can support the implementation of mHealth to improve vaccination rates and thus support planning and policy decisions on mHealth.

### Research questions

Our research questions are based on the population, intervention, comparator, and outcome [PICO] principles.

1. Do mHealth-supported children have a higher percentage of childhood vaccination than non-mHealth-supported children in Africa?

### Objectives

To summarize the effect of mHealth on childhood vaccination in Africa

To provide recommendations regarding mHealth in the African context

## Methods

### Study design

A systematic review and meta-analysis informed by Preferred Reporting Items for Systematic and Meta-analysis [PRISMA] standard guideline was employed.

### Inclusion criteria

The included have the following features:

1. Studies published from 2000 to 2023 were considered as mHealth is a newly emerging technology in Africa

2. Population: women with children aged <36 months from African countries

3. Interventions: used mHealth as an intervention [all types of service such as voice calls/message, video call/message, and text messages delivered through mobile phone]. The studies may or may not have other interventions to mHealth, and our study targets the mentioned interventions only.

4. Comparison: studies with groups of mothers who do not receive mHealth promotion to take and use the usual care or standard care without any interference

5. Outcome: primary endpoints of rate, coverage, timeliness, and vaccination completion compared with controls receiving standard care.

6. The secondary outcome was the difference among sample sizes, regions, and designs.

7. Designs: all trials, experimental and longitudinal cohort studies reported in the English language

## Exclusion criteria

Studies that

1. Lack of clarity in the method section [unknown design]

2. Are study protocols and review articles

3. Have a baseline difference between groups

## Search strategy and information source

We conducted a multi-step search strategy informed by the eligibility criteria outlined above. The databases included are CINAHL, EMBASE, PubMed, PsycINFO, Scopus, Web of Science, Google Scholar, Global Health, HINARI, and Cochrane Library. Additionally, the following source was explored: WHO website, mHealth Alliance, arxiv, AAS Open Research, Advance: a SAGE Preprints Community, AfricArxiv, AMRC Open Research, and International Development Research Centre reports, profit and nonprofit organizational websites, and the WHO International Clinical Trials Registry Portal [37].

**Key search terms.** Telehealth, telemedicine, telenursing, mobile app*, remote consultation, cellular phone, mHealth, wireless communication, mobile technology*, smartphone app*, vaccination, Immunization, child, and childhood

The search strategies for some databases were provided as a supplemental material [S1 File].

Two independent investigators who followed the PRISMA guidelines searched all articles that fulfilled the eligibility criteria.

## Study selection, quality appraisal, and data extraction

Endnote X20 was used to organize the articles and remove duplicates. After removing duplicates, the remaining papers were passed through title, abstracts, and full-text screening. A third observer decided on any disagreement between the two screening observers. The Joanna Briggs Institute [JBI] critical appraisal checklist was used to check the qualities of articles by informing eligibility criteria [38]. Two separate investigators conducted the final review and determined bias using the Cochrane tool [39], such as random sequence generation, allocation concealment, blinding of participants, blinding of personnel, blinding of outcome assessment, incomplete outcome data, selective reporting, and other sources of bias. Using this, all studies were classified as having low, unclear, or high bias. The author's name, year of publication,

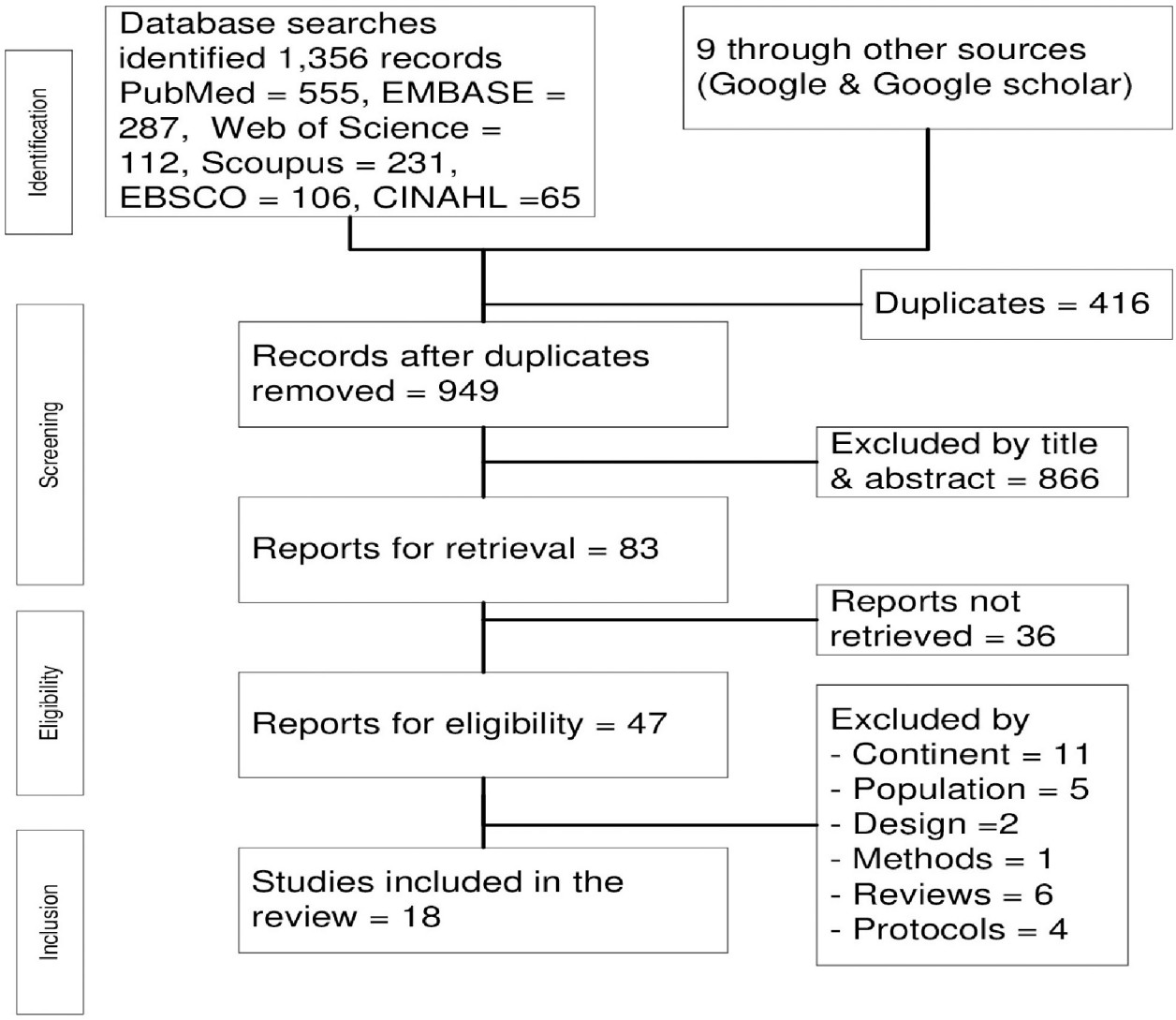

**Fig 1. The PRISMA diagram showing the methods of database screening.**

year of study, study design, study area, response rate, sample size, study quality score, and percentage were captured in a spreadsheet. The authors of the studies were contacted for any data that needed to be included.

We conducted database and manual searches between 23 January and 4th of March 2023. We found 1365 mother-children pair studies that were further assessed for eligibility criteria. Of these, 416 were excluded due to duplications. Then 949 studies were enrolled, and 866 were excluded by title or abstract. We excluded 36 studies by accessibility full-text. The included studies were further screened, and studies were additionally excluded due to ineligible continent [n = 11], population [n = 5], design [n = 2], methods [n = 1], reviews [n = 6], and protocols [n = 4] [Fig 1].

## Ethical approval and consent to participate

This study is a systematic review and metanalysis based on the protocol registered in PROPERO. The registration number is PROSPERO: CRD42023415956. Using these does not have

ethical risks. As part of a big project on the effect of mobile health on maternal and child health service uptaking, Arba Minch University IRB approved its awareness (AMU-IRB/1326/2022). All steps and activities are performed according to the available international guidelines.

## Statistical methods and analysis

The Revman 5.4.1, STATA 14, Comprehensive Meta-Analysis [CMA] interval prediction, and GRADEPro online software were applied for a detailed examination of the effect size, and the childhood vaccination data were presented using Forest Plots. The heterogeneity [$I^2$] and the between-studies variance [$\tau^2$] were quantified. We applied a p-value of less than 0.05 to declare associations [40]. The initial p<0.1 and $I^2$>50% decide the selection [random or fixed effect]. Sub-group analysis was conducted considering sample size, region, and design. The Egger regression asymmetry test and the Cochrane Collaboration Risk of Bias [CCRB] were used to check publication bias at p<0.05 [41], In contrast, missing studies were estimated using Duval and Tweedie's 'trim and fill method [42]. We used flow diagrams to display PRISMA steps, funnel plots to display publication biases, graphs to display the risk of bias, and forest plots to show effect size.

## Analyses of sensitivity

We checked the effect or influence of specific studies on the mean effect size and used a sub-group analysis to examine the observed heterogeneity. Model selection, risk of bias criteria, and loss to follow-up ≤25% were also considered [43].

## Results

### Characteristics of the included studies

Eighteen articles fulfilled the eligibility criteria out of 1,365 screened studies. Table 1 summarizes some characteristics of the included studies. The included studies account for a total sample size of 21070 [11246 mHealth and 9824 usual care], of which 11861 [6686 interventions and 5175 controls] used childhood vaccination [59.45% versus 52.67%]. Of 18 articles, only four are quasi-experimental studies [9, 44–46], while the remaining 14 are randomized controlled trials [11, 12, 16, 17, 20, 47–54]. Nigeria accounts for most studies [n = 8] [9, 18, 45, 46, 48, 51, 53, 54] followed by Kenya [n = 3] [11, 16, 47], Ethiopia [n = 2] [12, 50], Zimbabwe [n = 1] [20], South Africa [n = 1] [44], Ghana [n = 1] [49], Côte d'Ivoire [n = 1] [52], and Burkina Faso [n = 1] [17]. Nine studies were conducted in urban, seven in rural, and two in mixed settings [rural and urban]. The study population is mostly mother-child pairs except three caregivers included studies [Table 1] [11, 20, 54].

### Assessing the risk of bias

Fig 1 summarizes the quality assessment parameters of the included studies. Of 18 studies, 15 showed no selection bias [randomization and allocation], while two had high concerns and one had some concerns [9, 46]. Seven studies were rated to have high or some concerns regarding blinding of participants, researchers, and outcome assessors [11, 12, 16, 17, 44, 45, 48], five had incomplete outcome data biases [11, 20, 44, 51, 52], and four had other biases [Table 2] [17, 44, 46, 54] [Fig 2]. Based on the Cochrane risk of bias judgment tool, the risk of bias observed is presented [Fig 3].

### Childhood vaccination

Except for one study [50], all mHealth studies showed an improvement in childhood vaccination outcomes compared to usual care. Some studies had additional intervention arms, among

**Table 1. Characteristics of the studies included in the review.**

| study design | Population | Intervention | Outcome measure | Comparator |
|---|---|---|---|---|
| RCT | A community-based randomized control trial [RCT] was conducted in three woredas of Guraghe zone [Ezha, partial &Abeshge full intervention, Sodo Control] | The frontline SMS-based application was locally developed, customized, and integrated with the mobile phone system and the central server to send messages | % change in Immunization coverage [Penta1, Penta3, Measles]; | The usual care |
| RCT | Mothers or caregivers who recently delivered or during A third or seventh-day visit in Kadoma City Clinic in Mashonaland West province. Children <7 days | One-way SMS reminders sent 7 days, 3 days and 1 day before immunization appointment | Receipt of DPT-3 vaccines [coverage] and delay in immunization [timeliness] | The usual care [non SMS] |
| RCT | Children aged 0–3 months at recruitment paired with their mothers in a larger study in four local government areas[LGS] in Ibadan, Oyo state, Nigeria | Received one cellphone call reminder from the nurse/researcher two days before the child's next immunization appointment and a second call a day before the appointment date. | Coverage[percentage of full immunization] | The usual care |
| Quasi-Randomized Controlled Trial | Mothers–child pairs receiving ANC and PNC/EPI care in six public healthcare facilities in the Mobile Alliance for Maternal Action [MAMA] intervention in inner-city Johannesburg. Children <12 months | One-way maternal health SMS reminders sent twice weekly for each vaccination in the first year | Receipt of first-year infant vaccines [coverage] | The usual care |
| Quasi-Experiment | Mother-child pair that mothers attending immunizations clinics at 10 primary healthcare facilities in Kajola and Ibarapa north LGA in Oyo state | One-way SMS reminders sent 2day before and on the day of immunization appointment | Receipt of all infant vaccines [coverage] | The usual care |
| RCT | Mother-child pairs were recruited at all 29 health facilities at the time of the child's BCG immunization visit | Mothers were randomized to receive a voice or text reminder messages two days prior to the scheduled visit and two additional for missed doses | The mother/caretaker and child has to be resident of the Korhogo district and the mother had to have access to a functioning mobile phone | the usual care |
| RCT | Parturient women and their healthy newborn infants delivered at MCH-Ondo and Akure | Intervention group received a confirmatory text and call | Immunization completion at 12 month | the usual care |
| RCT | Caregiver–child pairs in eight health facilities in Egor LGA, Edo State Children due for first or second schedule of vaccines | One-way SMS reminders are sent 1 day before the immunization appointment. Follow-up messages were sent in cases of missed appointments | Receipt of infant vaccines [coverage] and timely receipt of vaccines [timeliness] | the usual care |
| RCT | The M-SIMU trial recruited HDSS village reporters to identify eligible caregivers and their infants in western Kenya. | Participants in the intervention groups received SMS reminders before scheduled pentavalent and measles immunization visits. | Proportion of fully immunized children | the usual care |
| RCT | Mother–child pairs in Langata, Machakos and Njoro districts. Children <6 weeks | One-way SMS reminders sent 2 days before and on the day of scheduled immunization day | Receipt of all infant vaccines [coverage] and timely receipt [timeliness] | the usual care |
| RCT | Mother-infant pairs presenting for the first vaccination appointment were randomized into four [three interventions, one control] groups, each consisting of 140 participants | Timeliness of appointments with reminders [calls or SMS], SMS health education | Vaccination completion rates | the usual care |
| RCT | The Mobile and Scalable Innovations for Measles Immunization [M-SIMI] study was a three-arm, individually randomized controlled trial conducted in Gem sub county, Siaya County, Kenya. | Two SMS reminders [SMS] sent 3 days, and 1 day before the scheduled MCV1 date, | MCV1 timely coverage | the usual care |
| RCT | A multi-centered randomized control trial was conducted at 33 primary health centers [phcs] in Lagos, Nigeria | Participants in the intervention group were sent an SMS text reminder two days before their appointments | The return rate for child vaccinations | the usual care |

(*Continued*)

**Table 1.** (Continued)

| study design | Population | Intervention | Outcome measure | Comparator |
|---|---|---|---|---|
| Cluster RCT | We conducted an open-label cluster randomized controlled trial with three arms in 15 communities [clusters] in Northern Ghana. | Communities were randomized to three groups: 1] a voice call reminder intervention [Intervention Group A, 5 communities]; 2] a CHV intervention with incentivized rewards [Intervention Group B, 5 communities]; 3] control [5 communities] | On-time completion of vaccinations | the usual care |
| RCT | Eligible mother-infant pairs from the University of Gondar Comprehensive Specialized Hospital and all the 8 health centers were included. | Participants assigned to the intervention group received mobile phone text message reminders one day before the scheduled vaccination visits | Primary outcomes of full and timely completion of vaccinations | the usual care |
| Quasi-Experiment | Mother–child pairs attending immunization clinics in Primary Health Centres in 14 lgas across six states and the Federal Capital Territory [FCT], Abuja. Children ≤2 months | Multiple one-way SMS reminders sent three times a week before the next immunization appointment | Receipt of all infant vaccines [coverage] and timely | the usual care |
| RCT | Mather-child pairs attending Center de sante et de promotion social[CSPS] in Colma 1 [medical district of do] | One-way SMS reminders before next due EPI vaccination session | Receipt of DPT-3 vaccine and timely receipt | the usual care |
| Quasi-Experiment | Quasi-experimental study were recent mothers of children not more than three weeks old selected from six local government areas of Kano State, Northern Nigeria | Mobile phone reminders [SMS and follow-up calls] were sent to mothers in the reminder group three days to and on the due date of their child's schedule | Penta 3 completion rate | the usual care |

**Table 2. Assessment of risk of bias parameters checked for each study.**

| Study | Random Sequence generation | Allocation concealment | Blinding of participants researcher/clinician | Blinding of outcome assessment | Incompleteness of data | Selective reporting | Other bias |
|---|---|---|---|---|---|---|---|
| Atnafu et al, 2017 | Yes | Yes | Yes | Yes | No | No | No |
| Bangure et al, 2015 | Yes | Yes | unknown | Unknown | No | no | No |
| Brown, et al, 2016 | Yes | Yes | Yes | Unknown | No | No | Unknown |
| Coleman et al, 2020 | Yes | Yes | No | Unknown | Yes | No | Yes |
| Dipeolu, 2017 | Yes | unknown | No | No | No | No | No |
| Dissieka et al, 2019 | Yes | Yes | Yes | Unknown | high | No | Unknown |
| Ekhaguere et al, 2019 | Yes | Yes | Yes | Yes | No | No | No |
| Eze et al, 2015 | Yes | yes | Yes | Yes | No | No | yes |
| Gibson et al, 2017 | Yes | Yes | Yes | No | high | No | No |
| Haji et al, 2016 | Yes | Yes | Unknown | Unknown | unknown | No | unknown |
| Ibraheem et al, 2021 | Yes | yes | Yes | Unknown | No | no | No |
| Kagucia et al, 2021 | Yes | Yes | No | Unknown | No | No | No |
| Kawakatsu et al, 2020 | Yes | Yes | Yes | No | No | No | No |
| Levine et al, 2021 | Yes | Yes | Yes | Yes | No | no | No |
| Mekonnen et al, 2021 | Yes | Yes | Unknown | Yes | unknown | No | No |
| Oladepo et al, 2021 | no | unknown | unknown | Unknown | unknown | No | No |
| Schlumberger et al, 2015 | Yes | Yes | Yes | No | unknown | No | Yes |
| Yunusa et al, 2022 | No | No | Yes | Yes | No | No | Yes |

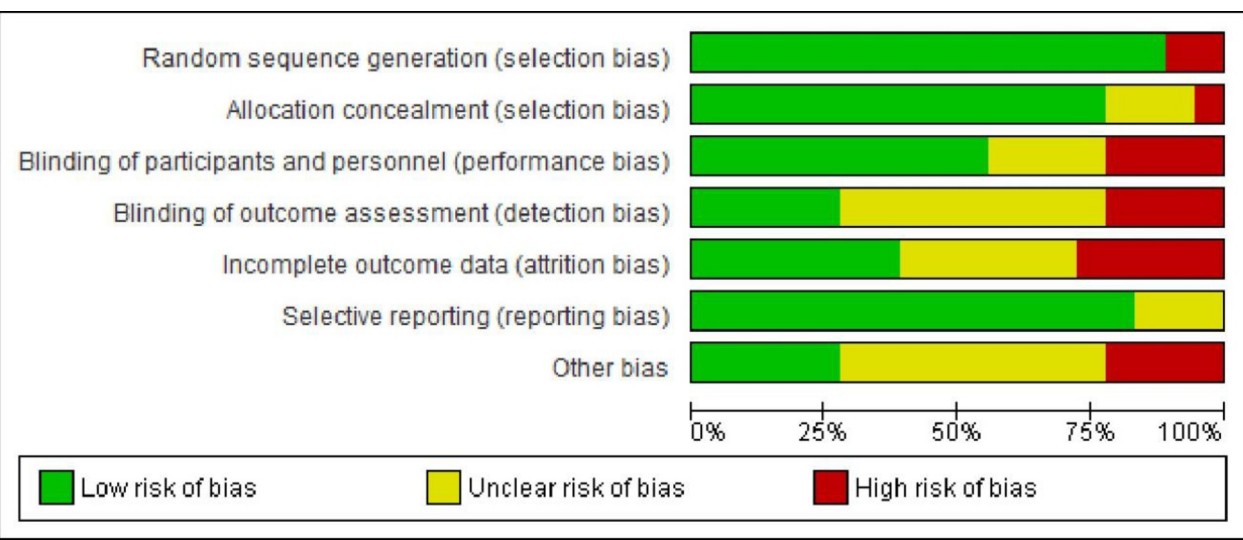

**Fig 2. Risk of bias graph presented as percentages across all included studies using Cochrane assessment criteria.**

which three had incentive intervention arms [11, 16, 49]. In all cases, monetary incentives groups had significantly better vaccination rates than SMS alone. However, the SMS intervention indicated pentavalent, polio, and measles vaccination improvement. The average effect size using the random-effects model [OR = 2.15 [95% CI: 1.70–2.72; P<0.001; $I^2$ = 90%; RD = 0.11, 95%CI: 0.07–0.16]. The confidence interval of this effect size does not perfectly overlap the actual effect size [0.81, 5.74] [Fig 4], which might show some uncertainty in the estimation of the effect size [Fig 4]. Despite these variations, mHealth has been shown to significantly improve childhood vaccination completion compared to standard of care [Fig 5].

## Sensitivity test

The mean heterogeneity [$I^2$] of 90% and $tau^2$ of 0.20 indicate that the effect size significantly varies. A one-by-one exclusion sensitivity analysis shows that the results are stable, and no study causes an excessive change by its presence or absence in the analysis and does not influence the effect size significantly. An examination of the student zed residuals revealed that none of the studies had a value greater than ± 2.9913, and our review does not indicate outliers. According to Cook's distances, none of the studies could be considered overly influential. Neither the rank correlation nor the regression test showed any funnel plot asymmetry [Fig 6]. The Harbord test also suggested the absence of publication bias [*P* = 0.591].

## Certainty of the evidence

We checked the quality of evidence for this systematic review and meta-analysis with the GRADEpro online application on a four-grade scale of very low, low, moderate, and high [Fig 7]. The overall quality of the body of evidence is found to be reasonable [Fig 8]. The meta-regression was conducted to see the impact of variables on the pooled effect size. The effect of each sample size and follow-up time were regressed on the effect size and were not statistically significant in indicating the reverse relationships [p = 0.947 for follow-up time and p = 0.960].

## Subgroup analysis

**Analysis by regions.** A subgroup analysis by African regions [Western, Southern, and Easter] indicates that there is a significant effect on the application of mHealth to the

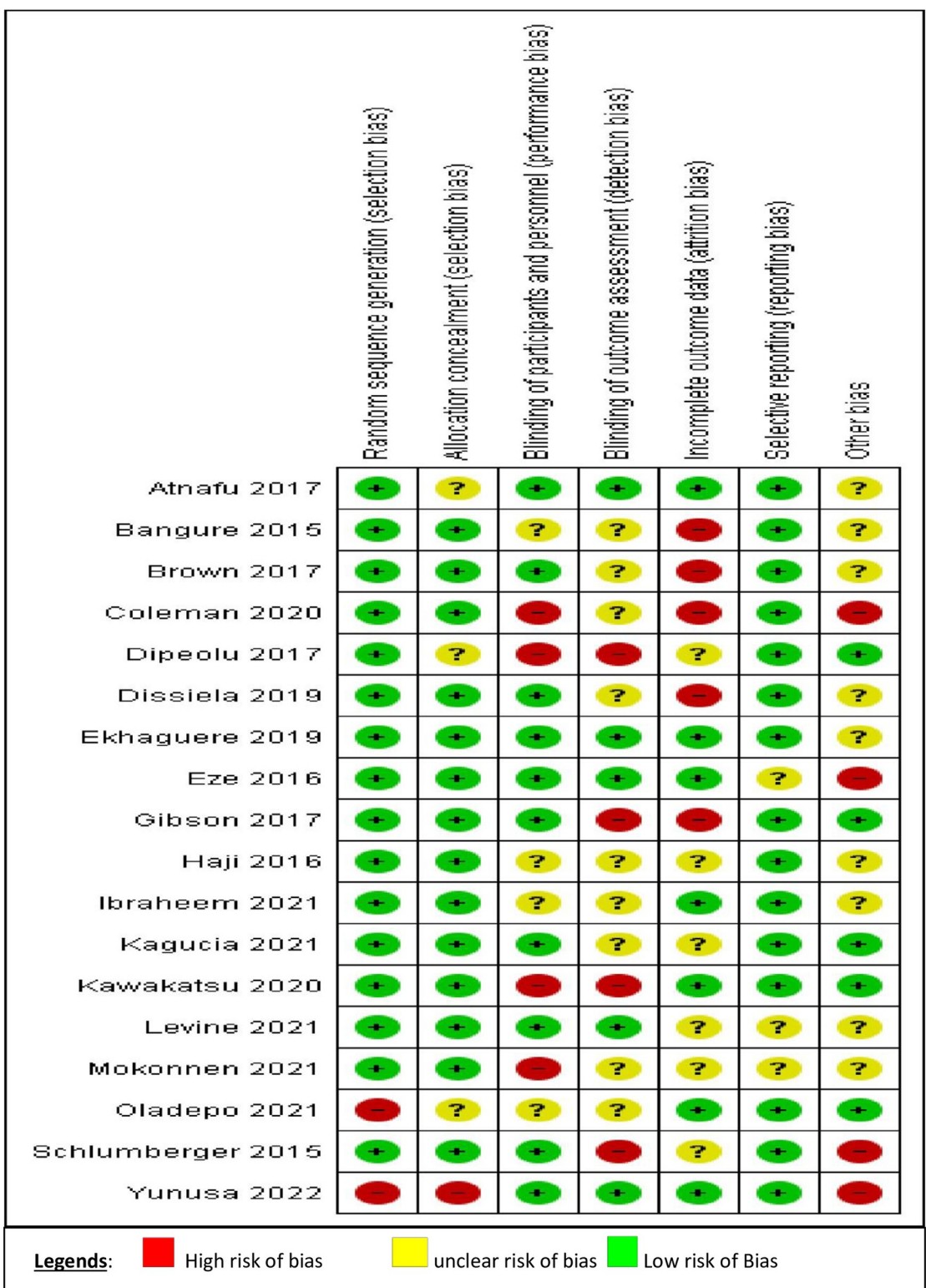

**Fig 3. Risk of bias authors' judgments summary applying Cochrane risk of bias criteria.**

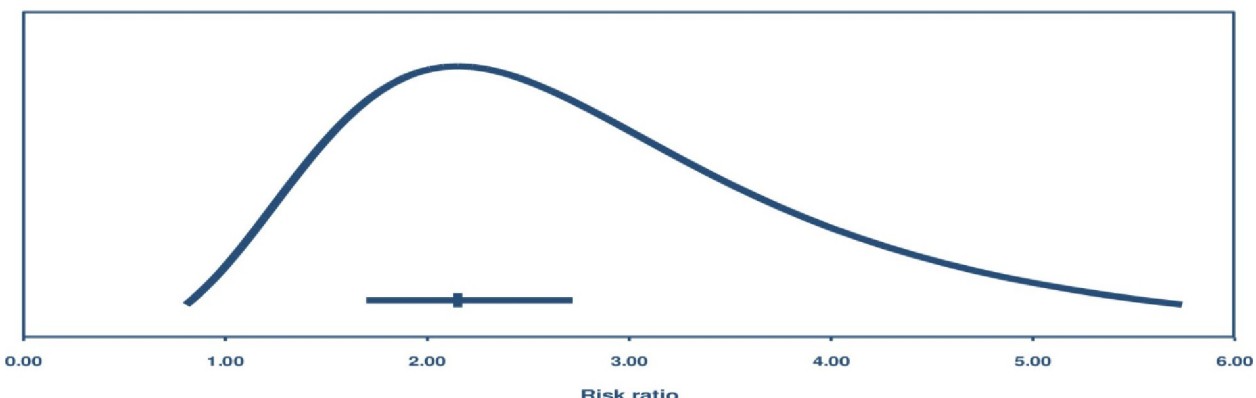

**Fig 4. The distribution of true effect size among the comparable universal populations.**

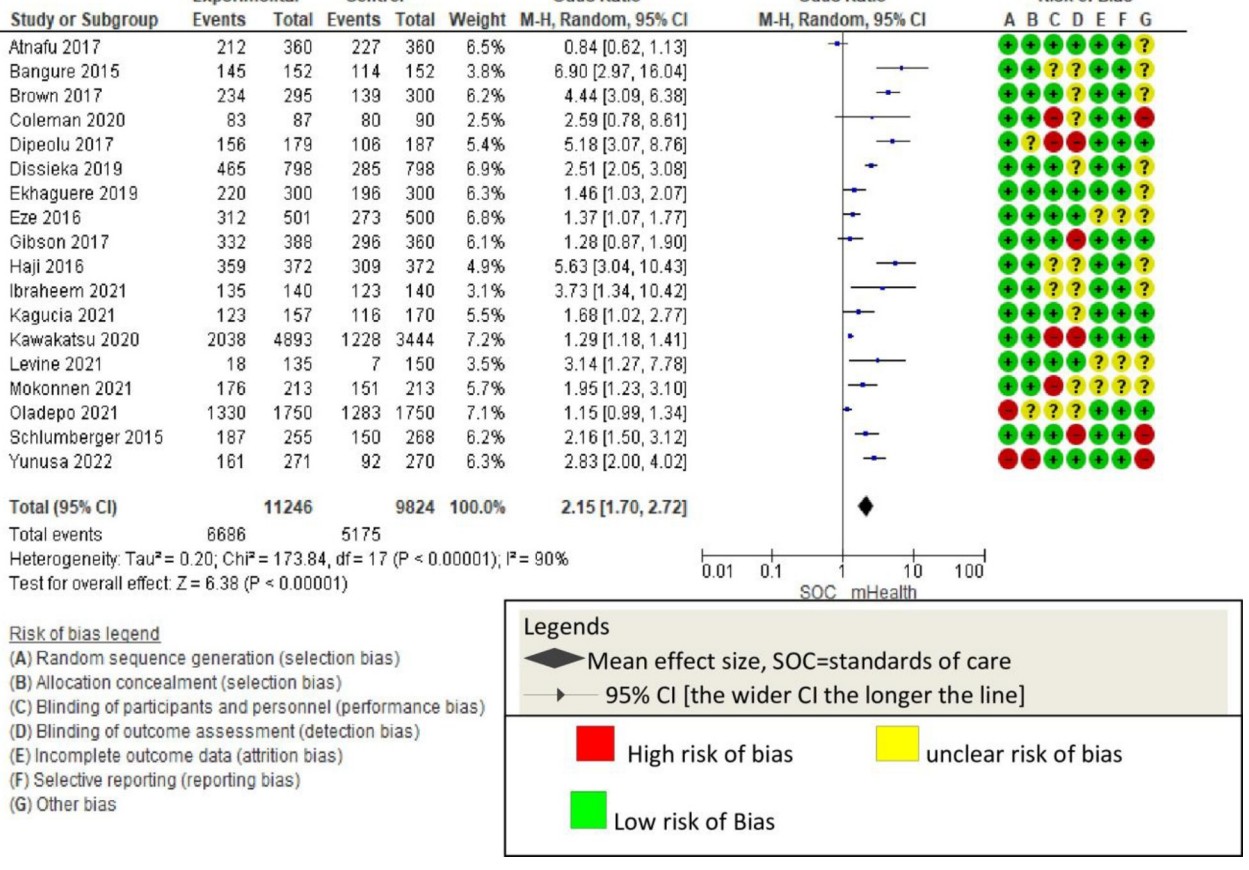

**Fig 5. The forest plot displaying overall effect size and relation among the 18 studies.**

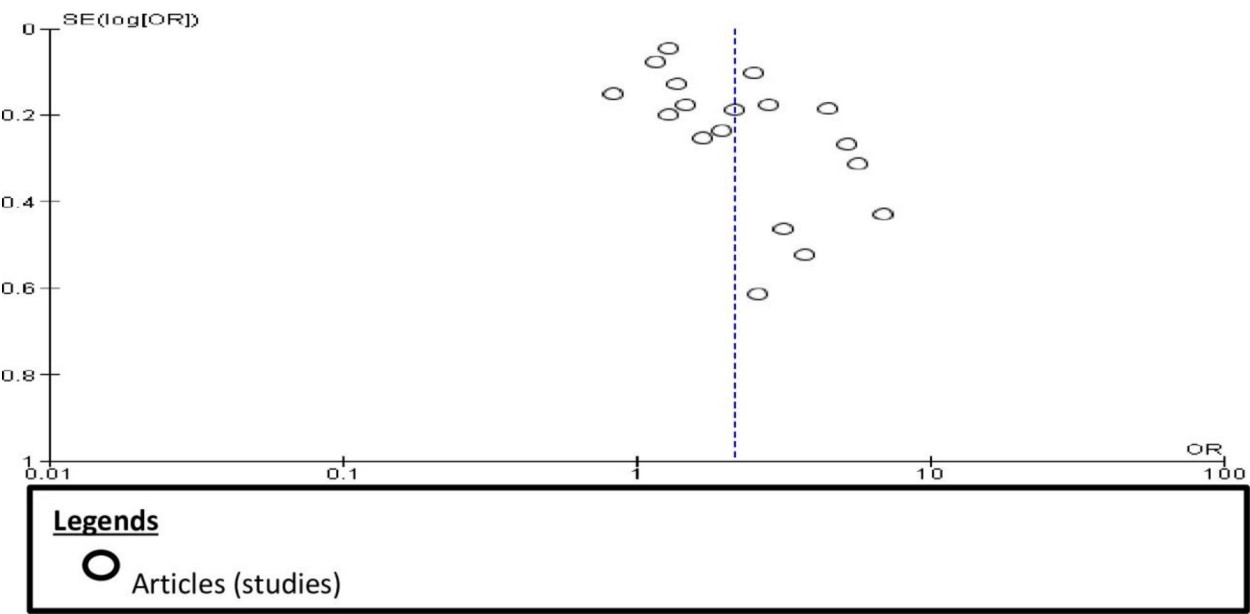

**Fig 6. A funnel plot showing the symmetry of studies included in the review.**

improvement of childhood vaccination [OR = 2.15; 95%CI: 1.70–2.72; p = 0.22; $I^2$ = 34.7%]. It showed that regional variation is a source of outcome heterogeneity [Fig 9].

Eight studies reported pentavalent-3 vaccine as an outcome and showed a significant pentavalent-3 vaccination improvement among children [OR = 2.21, 95% CI: 1.62–3.02; P<0.002; $I^2$ = 81%; RD = 0.11, 95%CI: 0.07–0.15] with variation among outcomes [Fig 10]. It has a higher risk difference [RD] that is equal to the RD of the overall analysis. This shows a high variation in pentavalent outcomes [Fig 10].

Three studies reported polio vaccination and showed a significant change in mHealth intervention groups compared to usual care [OR = 2.27; 95%CI: 1.51–3.40; p<0.30; $I^2$ = 18] [Fig 11].

Six studies also reported Measles vaccination as an outcome and showed a significant improvement through the mHealth intervention [OR = 2.19, 95% CI: 1.63–2.96; P<0.001; $I^2$ = 80%] [Fig 12].

Additionally, three studies used incentive [monetary] and reported significant improvement in childhood vaccination [OR = 1.59, 95% CI: 1.22–2.08; P<0.47; $I^2$ = 0%] compared to controls and SMS-only intervention [Fig 13].

## Discussion

This review showed that the overall childhood vaccination is 2.15 times higher among mHealth-used groups. Eight studies sought the effect of mHealth on pentavalent [penta-3] vaccination in African countries [9, 11, 12, 20, 46, 47, 52, 53]. All these studies reported a significant improvement in penta-3 vaccination rates positively. Previous systematic reviews reported consistent findings in childhood penta-3 vaccination [55, 56]. This evidence suggests that mHealth implementation should be considered to improve pentavalent vaccination in African countries [Fig 10]. Four studies with poliovirus vaccination outcomes reported that mHealth interventions significantly improved childhood polio vaccination rates [9, 20, 44, 49]. Other evidence shows that the mHealth intervention improves polio vaccination rates along with other vaccinations and indicates the potential success of mHealth implementation [55–

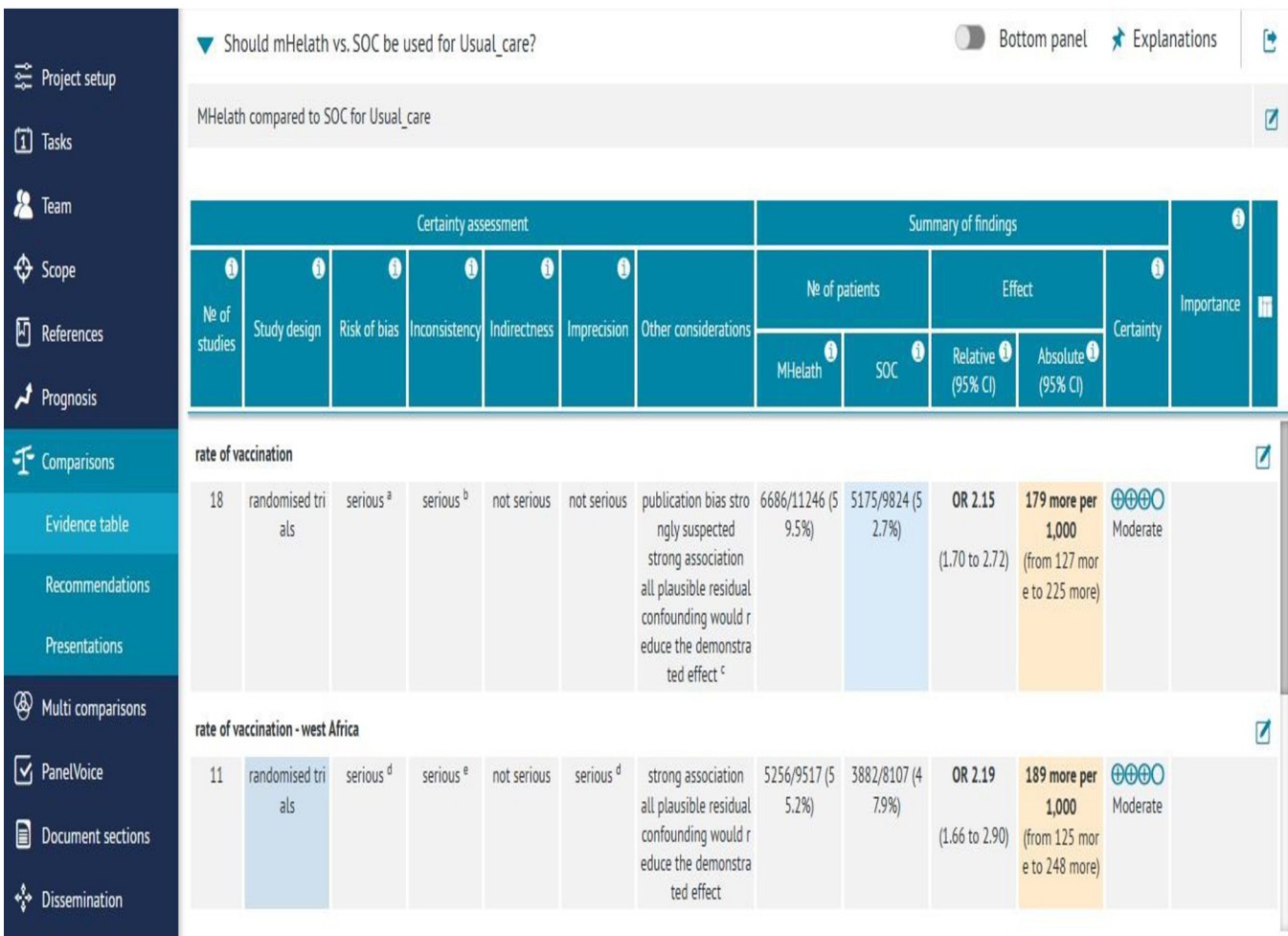

**Fig 7. A GRADEPro showing the certainty of the evidence.**

57] [Fig 11]. The six studies that reported mHealth application to improve measles vaccination showed a significant improvement in measles vaccination rates [9, 11, 12, 16, 52, 53]. Other systematic reviews and meta-analysis studies also showed the potential of mHealth in improving measles vaccination [55, 56]. Overall, childhood measles vaccinations were enhanced with the application of mobile technology in Africa [Fig 12].

In addition to reminders, three studies [11, 16, 49] sought improvement in childhood vaccination using monetary incentives. In all cases, financial incentives improved vaccination revisits among the mothers who received economic incentives for their last visits compared to the usual care. Although there is no previous systematic review evidence on financial-based studies, the current information indicates that this will improve childhood vaccination rates [Fig 13]. Using mHealth was stated to be cheaper, cost-saving, and life-saving mechanism [58]. Evidence suggests that incentives might provide an additional dimension to mHealth to get more effect [11]. However, applying low-cost simple mobile phones to use mHealth has been as effective as using incentives while incentives are additional costs for the governments and might not be sustainable in low-economy regions [59].

Overall, the timeliness, rate, and dropout rate of vaccinations improved through mHealth assistance. However, there is a challenging heterogeneity in the observed effect size, which led

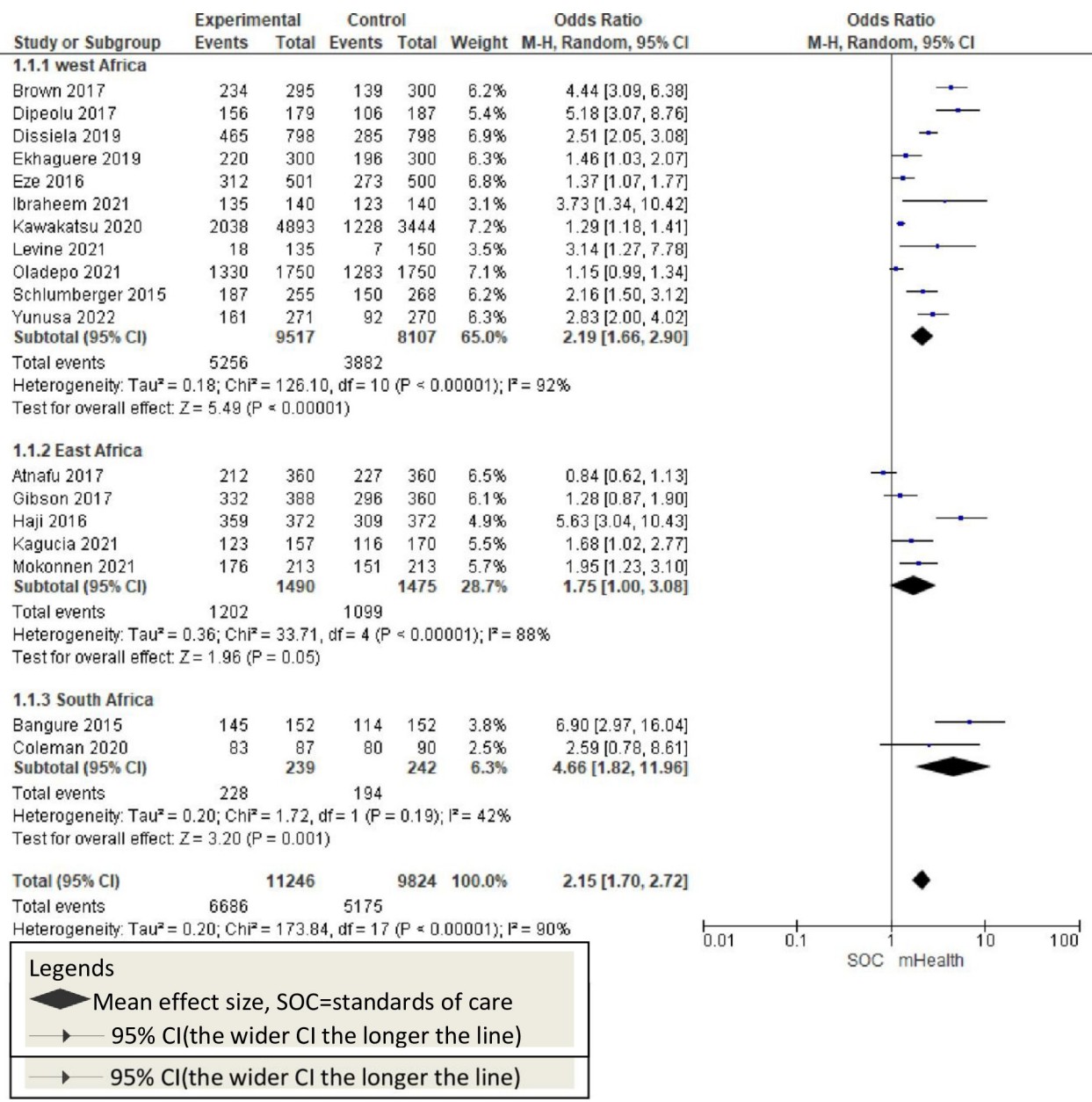

**Fig 8. Forest plot of pooled childhood vaccination by regions in Africa.**

us to examine settings, sample size, and varying duration of experiments. Socio-demographic challenges such as educational status and living conditions may also contributed to heterogeneity [54, 60]. The subgroup analysis by region showed that there is a substantial variation in effect size among regions. All diagnostic methods applied to identify the heterogeneity showed that none of the included studies are overly influential and underlines the stability of the findings. One of the studies among the 18 appeared to have a contradicting finding after text reminder intervention in Ethiopia [50]. However, examining the evidence showed that mothers have many reasons for reduced visits for childhood vaccination, including waiting time, as caregivers are too busy, the absence of health professionals to provide vaccination on specific days, and the lack of the vaccine itself on some occasions. This indicates that the study in

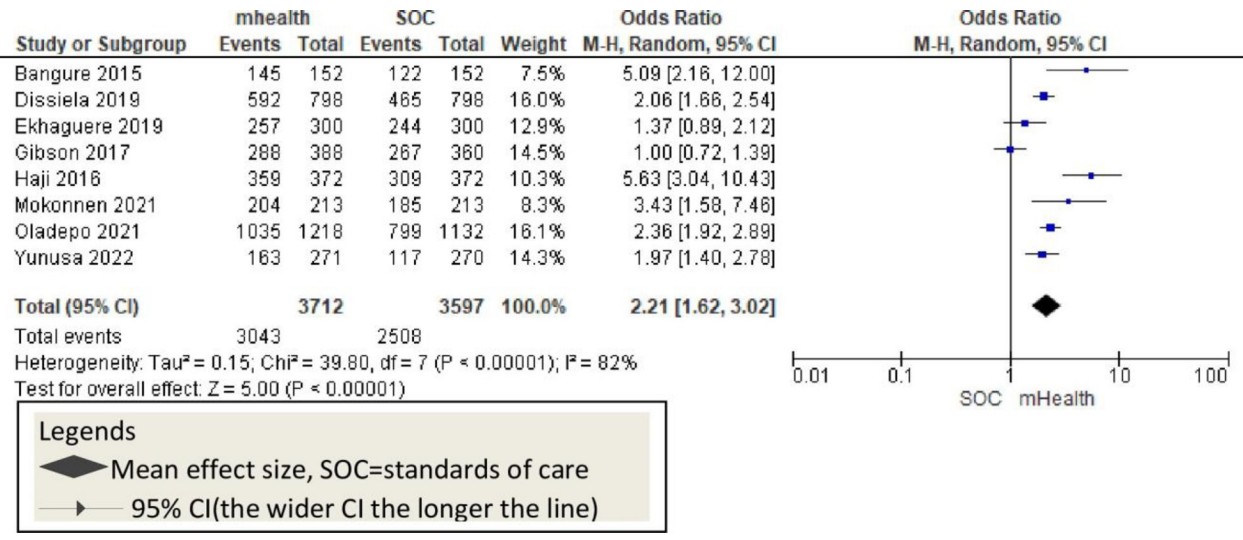

**Fig 9. Forest plot of pooled pentavalent-3 vaccination coverage.**

Ethiopia was conducted in a more challenging environment. It was also noticed that resource limitations such as the limited number of service providers and limited access to vaccine resources could still hinder childhood vaccination even in the presence of successful simple technologies such as mHealth.

The effect of mHealth on childhood vaccination is among the winning ideas throughout the globe especially, in developing countries. There is a positive relationship between childhood vaccination and mobile health applications globally [21, 32, 58, 61, 62]. Africa as one of the underdeveloped continents has the highest rate of unvaccinated children. A new UNICEF report shows that 12.7 million children were under-vaccinated in 2021, including 8.7 million who never started vaccination [zero-dose children] [63]. The application of mHealth has shown a potential implication of increasing childhood vaccination in the African continent [8, 14, 64]. The lack of summarized evidence on the effect of mHealth on childhood vaccination in the continent was covered in this review. Thus, this review is current, and comprehensive to include previous, and the most current evidence to assist planning and policy decisions.

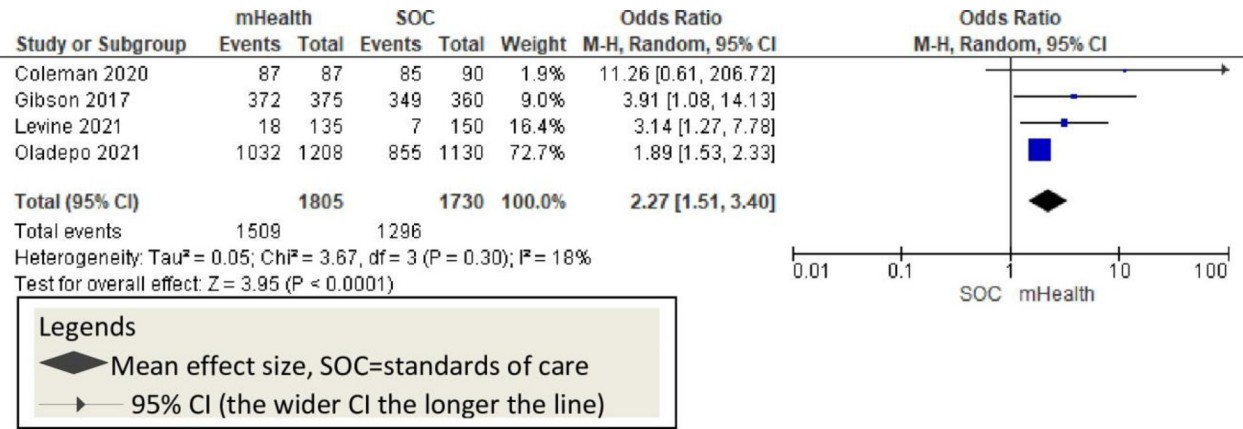

**Fig 10. Forest plot of pooled polio vaccination coverage.**

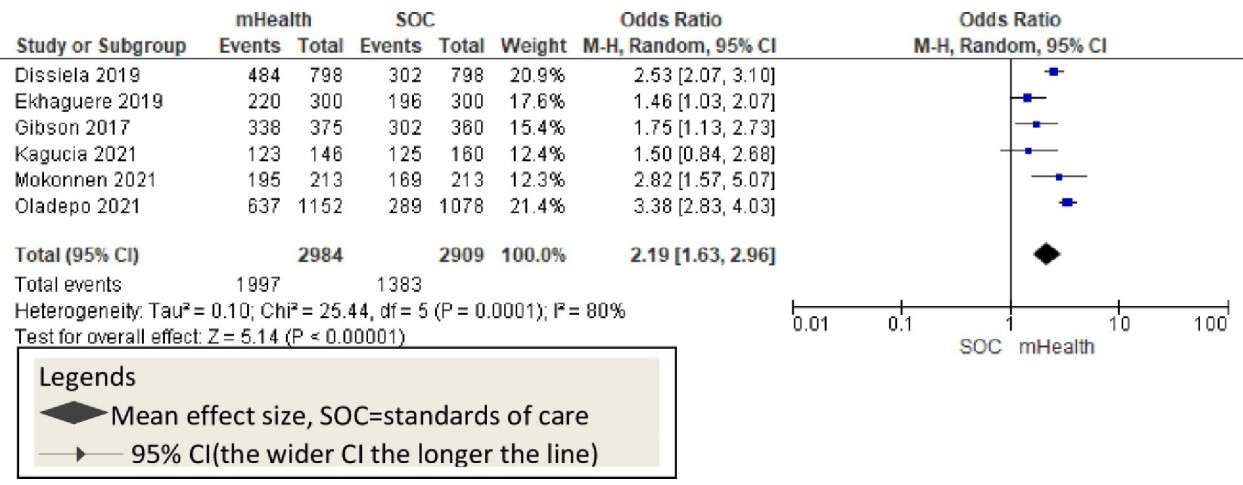

**Fig 11. Forest plot of pooled Measles vaccination coverage.**

## The strengths and limitations of this study

This review's main strength is extracting, presenting, and evaluating the evidence using standard guidelines. Publication bias, quality of evidence, and source of variation of outcomes were addressed using the guidelines. However, there are still some limitations. Many studies are from Western and Eastern Africa, while the other regions contribute markedly fewer studies. Additional studies have either an unclear or a high risk of bias, such as blinding participants, researchers, data collectors, and loss of follow-up. This increases the risk of overall biases in outcome evaluation. However, GRADEPro [Fig 7], which we used to examine the quality of the body, considered the existence of those issues and concluded that bias is not the concern.

## Conclusions

The application of mHealth increased childhood vaccination by more than double among participants in the included studies in Africa. This shows a high potential of improving childhood vaccination in the African continent. Although there are some variations based on the level of development of regions and the inclusion of additional incentives in some studies, it is still

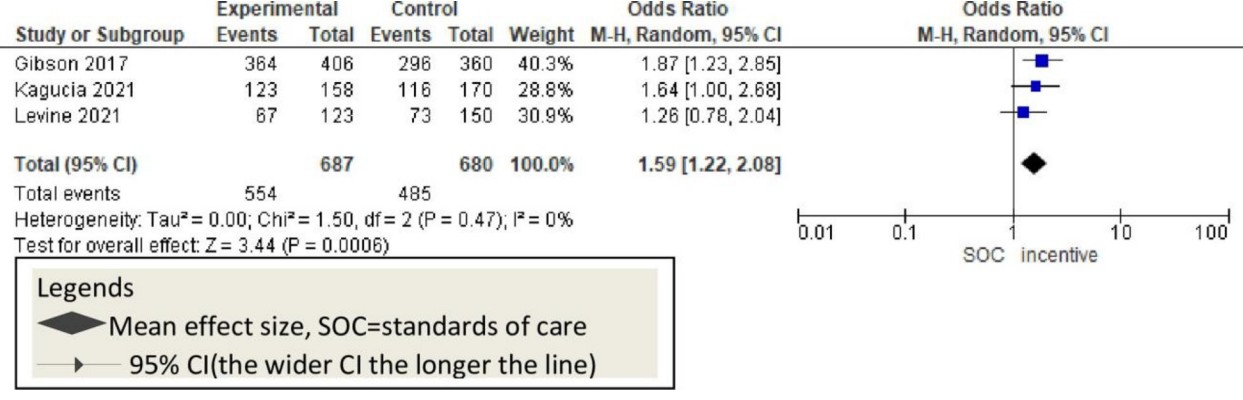

**Fig 12. Forest plot of the pooled effect of incentive on vaccination timeliness.**

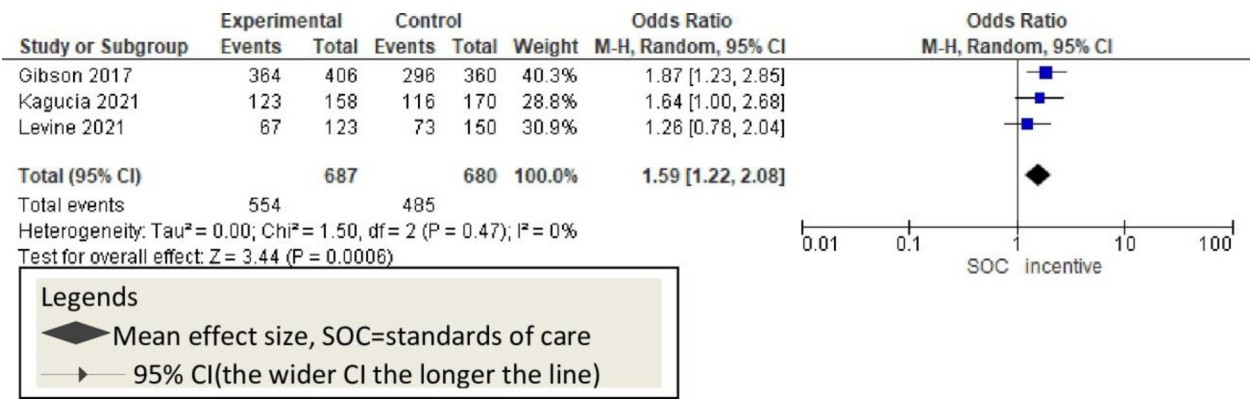

**Fig 13. Forest plot of the pooled effect of incentive on vaccination timeliness.**

productive in improving childhood vaccination in all regions. It improved coverage, timeliness, and completeness and reduces the dropout rates aspect of the vaccination. Based on this evidence the Global Vaccine Action Plan, which states all individuals and communities should enjoy life that is free from vaccine-preventable diseases [65], can be achieved by applying mHealth in Africa. This is also an opportunity that the WHO African region plans, which focus on increasing vaccination coverage, interrupting poliovirus transmission, eliminating measles, and controlling other vaccine-preventable diseases [66], can be achieved by applying mHealth. Additionally, this review contains conclusive evidence that can support continental and international efforts to use mHealth to improve vaccination.

## Supporting information

**S1 Checklist. PRISMA 2020 checklist.**
(DOCX)

**S1 File. Search strategies.**
(DOCX)

**S1 Data. Extracted data.**
(XLSX)

## Acknowledgments

The authors are thankful to Maastricht University for providing support to access various databases in the review process.

## Author Contributions

**Conceptualization:** Girma Gilano.

**Data curation:** Sewunet Sako.

**Formal analysis:** Girma Gilano, Berihun Molla.

**Investigation:** Girma Gilano.

**Methodology:** Girma Gilano, Sewunet Sako, Berihun Molla, Andre Dekker, Rianne Fijten.

**Project administration:** Sewunet Sako.

**Software:** Girma Gilano, Berihun Molla, Rianne Fijten.

**Supervision:** Andre Dekker, Rianne Fijten.

**Validation:** Andre Dekker, Rianne Fijten.

**Visualization:** Rianne Fijten.

**Writing – original draft:** Girma Gilano, Sewunet Sako, Andre Dekker, Rianne Fijten.

**Writing – review & editing:** Girma Gilano, Sewunet Sako, Andre Dekker, Rianne Fijten.

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
