## [Decision Letter · Decision Letter 0]

20 Sep 2023

PONE-D-23-16959The effect of mHealth on childhood vaccination in Africa: a systematic review and Meta-analysisPLOS ONE

Dear Dr. Gilano,

Thank you for submitting your manuscript to PLOS ONE. After careful consideration, we feel that it has merit but does not fully meet PLOS ONE’s publication criteria as it currently stands. Therefore, we invite you to submit a revised version of the manuscript that addresses the points raised during the review process.

We look forward to receiving your revised manuscript.

Kind regards,

Olushayo Oluseun Olu

Academic Editor

PLOS ONE

Journal Requirements:

4. We are unable to open your Supporting Information file supporting files.rar. Please kindly revise as necessary and re-upload.

Reviewers' comments:

Reviewer's Responses to Questions

**Comments to the Author**

1. Is the manuscript technically sound, and do the data support the conclusions?

Reviewer #1: Yes

Reviewer #2: Yes

2. Has the statistical analysis been performed appropriately and rigorously? 

Reviewer #1: Yes

Reviewer #2: Yes

3. Have the authors made all data underlying the findings in their manuscript fully available?

Reviewer #1: Yes

Reviewer #2: Yes

4. Is the manuscript presented in an intelligible fashion and written in standard English?

Reviewer #1: Yes

Reviewer #2: Yes

5. Review Comments to the Author

Reviewer #1: Well written paper but will like the conclusion to state clearly by how much mhealth has improved vaccination and which aspect of mhealth is the most effective, as this can guide MOH on where to focus their energies. The fact that economic motivation also increased coverage, can yo justify why not use the scarce resources this way rather than on mhealth.

Reviewer #2: Reviewer Report: PONE-D-23-16959

The effect of mHealth on childhood vaccination in Africa: a systematic review and

Meta-analysis

Date: 05/10/2018

Gilano and colleagues appreciate the significant role that mHealth could play in combating Vaccine Preventable Diseases by improving access to health information and decision making on vaccinations. The authors attempt to provide more evidence to substantiate the effect of mHealth on childhood vaccination in Africa by way of a systematic review and Meta-analysis. The study is timely especially as it provides essential information to assist planning and policy decisions pertaining to use of mHealth for the role-out and uptake of vaccines. The study is robust enough and uses the recommended methods and the authors are commended for expending efforts towards provision of evidence on this subject. I recommend the publication of this manuscript albeit with minor and major concerns to be addressed as outlined below.

Minor Revisions

In the Abstract, in the third sentence under introduction, the term “the” before mHealth should be dropped to read as “……. information to support mHealth introduction.” The lst sentence should be revised to read as “This study provides essential information to assist planning and policy decisions regarding the use of mHealth for childhood vaccination.”

Introduction in the main body: In the second paragraph the sentence “However, vaccination coverage is decreasing currently and will be only 81% in 2021 throughout the globe (4).” Should read as “However, global vaccination coverage is decreasing and was only 81% in 2021 (4).” In paragraph 4 the sentence “Additionally, a mobile phone-based platform improved the childhood vaccination rate in Kenya (14)” should end with a full stop. Paragraph 7 has a different font size compared to the rest of the text in the introduction chapter.

Major Revisions

Results: The results section has sentences and sections explaining the procedures that ideally should be part of the methodology.

Discussion: This section could benefit from rearrangement and strengthening. In the current state, it is sparingly indicating the main findings and minimally relating them to some studies done elsewhere throughout the section. However, a more logical presentation would markedly improve the paper in the lines of: Presenting the findings in the first paragraph or two and how they corroborate other local studies. Followed by the challenges experienced during the study. Then dwelling on how the current findings relate to other activities internationally. Then proceeding on to give any recommendations that could inform similar work or strengthen policy implementation. Finally crown it all in the conclusion.

Conclusion: This section would also befit from strengthening by reinforcing key findings and implications.

6. PLOS authors have the option to publish the peer review history of their article (what does this mean?). If published, this will include your full peer review and any attached files.

Reviewer #1: **Yes: **Sylvester Maleghemi

Reviewer #2: No

---

## [Author Response · Author response to Decision Letter 0]

13 Oct 2023

The authors are very grateful to the editors and reviewers for their time and their great attitude to make the right information reach reader in the right language and method. This is a privilege we got by being part of the scientific community. There is a variety of questions asked from many perspectives. We tried to compile the responses in relation to each comment from all authors, which mean the answers may not be straightforward in some of the cases. The document went under complete editing and reading where the track changed version can tell the story. In a few cases, we may not understand and address some comments. In that case, we are happy to comply any further comments. All the responses were pasted following the original comments by reviewers and editors as follow.

Editorial comments

Comments: Please ensure that your manuscript meets PLOS ONE's style requirements, including those for file naming. The PLOS ONE style templates can be found at 

Response: Thank you for the comment. We improved the style as per the PLOS ONE's style requirements

Comments: In your Data Availability statement, you have not specified where the minimal data set underlying the results described in your manuscript can be found. PLOS defines a study's minimal data set as the underlying data used to reach the conclusions drawn in the manuscript and any additional data required to replicate the reported study findings in their entirety. All PLOS journals require that the minimal data set be made fully available. For more information about our data policy, please see http://journals.plos.org/plosone/s/data-availability.

Response: Thank you for the comment. we uploaded data used for this review as an additional file

Comments: Upon re-submitting your revised manuscript, please upload your study’s minimal underlying data set as either Supporting Information files or to a stable, public repository and include the relevant URLs, DOIs, or accession numbers within your revised cover letter. For a list of acceptable repositories, please see http://journals.plos.org/plosone/s/data-availability#loc-recommended-repositories. Any potentially identifying patient information must be fully anonymized.

Response: Thank you for the comment. We uploaded data used for this review as an additional file

Comments: Important: If there are ethical or legal restrictions to sharing your data publicly, please explain these restrictions in detail. Please see our guidelines for more information on what we consider unacceptable restrictions to publicly sharing data: http://journals.plos.org/plosone/s/data-availability#loc-unacceptable-data-access-restrictions. Note that it is not acceptable for the authors to be the sole named individuals responsible for ensuring data access.

Responses: Thank you for the comment. There are no ethical restrictions and we uploaded data used for this review as an additional file

Comments: Please include captions for your Supporting Information files at the end of your manuscript, and update any in-text citations to match accordingly. Please see our Supporting Information guidelines for more information: http://journals.plos.org/plosone/s/supporting-information. 

Response: Thank you for the comment. We included captions for Supporting Information files at the end of the document

Comments: We are unable to open your Supporting Information file supporting files.rar. Please kindly revise as necessary and re-upload.

Responses: Thank you for the comment. We solved this by combining files in a single doc and provided as a supporting information

Reviewer#1

Comments: Well written paper but will like the conclusion to state clearly by how much mHealth has improved vaccination and which aspect of mHealth is the most effective, as this can guide MOH on where to focus their energies. 

Responses: Thank you for the comment. Both conclusions (abstract and end) are formatted as per comments(page 2, line 49-53, page 14, line 341-345)

Comments: The fact that economic motivation also increased coverage, can you justify why not use the scarce resources this way rather than on mHealth.

Responses: Thank you for the comment. In addition to including this information in conclusion, it also included in discussion section (page 13, line 316-320)

Reviewer#2 

Comments: In the Abstract, in the third sentence under introduction, the term “the” before mHealth should be dropped to read as “……. information to support mHealth introduction.” The lst sentence should be revised to read as “This study provides essential information to assist planning and policy decisions regarding the use of mHealth for childhood vaccination.”

Responses: Thank for the comments. We improved introduction section of abstract as per comments (page 2, line 31-35)

comments: Introduction in the main body: In the second paragraph the sentence “However, vaccination coverage is decreasing currently and will be only 81% in 2021 throughout the globe (4).” Should read as “However, global vaccination coverage is decreasing and was only 81% in 2021 (4).” In paragraph 4 the sentence “Additionally, a mobile phone-based platform improved the childhood vaccination rate in Kenya (14)” should end with a full stop. Paragraph 7 has a different font size compared to the rest of the text in the introduction chapter.

Response: Thank for the comments. All the comments were incorporated.

Comments: Results: The results section has sentences and sections explaining the procedures that ideally should be part of the methodology.

Responses: Thank for the comments. We moved the section that describing PRISMA that is accidently presented under result section to Method section before PRISMA diagram.

Comments: Discussion: This section could benefit from rearrangement and strengthening. In the current state, it is sparingly indicating the main findings and minimally relating them to some studies done elsewhere throughout the section. However, a more logical presentation would markedly improve the paper in the lines of: Presenting the findings in the first paragraph or two and how they corroborate other local studies. Followed by the challenges experienced during the study. Then dwelling on how the current findings relate to other activities internationally. Then proceeding on to give any recommendations that could inform similar work or strengthen policy implementation. Finally crown it all in the conclusion.

Responses: Thank for the comments. The discussion section is modified as per the comments (page 12-14, line 275-326)

Comments: Conclusion: This section would also befit from strengthening by reinforcing key finding and implications.

Responses: Thank for the comments. The conclusion section in abstract and end of the document was modified and strengthened in response to both reviewers’ comments (page 2, line 49-53 and page 14-15, line 339-351)

---

## [Editor Report · Decision Letter 1]

2 Nov 2023

The effect of mHealth on childhood vaccination in Africa: a systematic review and Meta-analysis

PONE-D-23-16959R1

Dear Dr. Gilano,

We’re pleased to inform you that your manuscript has been judged scientifically suitable for publication and will be formally accepted for publication once it meets all outstanding technical requirements.

Kind regards,

Olushayo Oluseun Olu

Academic Editor

PLOS ONE
---

## [Editor Report · Acceptance letter]

8 Nov 2023

PONE-D-23-16959R1 

The effect of mHealth on childhood vaccination in Africa: a systematic review and Meta-analysis 

Dear Dr. Gilano:

I'm pleased to inform you that your manuscript has been deemed suitable for publication in PLOS ONE. Congratulations! Your manuscript is now with our production department. 

Kind regards, 

on behalf of

Dr. Olushayo Oluseun Olu 

Academic Editor

PLOS ONE